# Real-Time Determination of Intracellular cAMP Reveals Functional Coupling of G_s_ Protein to the Melatonin MT_1_ Receptor

**DOI:** 10.3390/ijms25052919

**Published:** 2024-03-02

**Authors:** Lap Hang Tse, Suet Ting Cheung, Seayoung Lee, Yung Hou Wong

**Affiliations:** 1Division of Life Science and the Biotechnology Research Institute, Hong Kong University of Science and Technology, Hong Kong, China; lhtseaa@connect.ust.hk (L.H.T.); anniecst@ust.hk (S.T.C.); sleecr@connect.ust.hk (S.L.); 2State Key Laboratory of Molecular Neuroscience, Molecular Neuroscience Center, Hong Kong University of Science and Technology, Hong Kong, China; 3Hong Kong Center for Neurodegenerative Diseases, 17 Science Park West Avenue, Hong Kong Science Park, Shatin, Hong Kong, China

**Keywords:** adenylyl cyclase, G protein, melatonin receptor, modeling

## Abstract

Melatonin is a neuroendocrine hormone that regulates the circadian rhythm and many other physiological processes. Its functions are primarily exerted through two subtypes of human melatonin receptors, termed melatonin type-1 (MT_1_) and type-2 (MT_2_) receptors. Both MT_1_ and MT_2_ receptors are generally classified as G_i_-coupled receptors owing to their well-recognized ability to inhibit cAMP accumulation in cells. However, it remains an enigma as to why melatonin stimulates cAMP production in a number of cell types that express the MT_1_ receptor. To address if MT_1_ can dually couple to G_s_ and G_i_ proteins, we employed a highly sensitive luminescent biosensor (GloSensor^TM^) to monitor the real-time changes in the intracellular cAMP level in intact live HEK293 cells that express MT_1_ and/or MT_2_. Our results demonstrate that the activation of MT_1_, but not MT_2_, leads to a robust enhancement on the forskolin-stimulated cAMP formation. In contrast, the activation of either MT_1_ or MT_2_ inhibited cAMP synthesis driven by the activation of the G_s_-coupled β_2_-adrenergic receptor, which is consistent with a typical G_i_-mediated response. The co-expression of MT_1_ with G_s_ enabled melatonin itself to stimulate cAMP production, indicating a productive coupling between MT_1_ and G_s_. The possible existence of a MT_1_-G_s_ complex was supported through molecular modeling as the predicted complex exhibited structural and thermodynamic characteristics that are comparable to that of MT_1_-G_i_. Taken together, our data reveal that MT_1_, but not MT_2_, can dually couple to G_s_ and G_i_ proteins, thereby enabling the bi-directional regulation of adenylyl cyclase to differentially modulate cAMP levels in cells that express different complements of MT_1_, MT_2_, and G proteins.

## 1. Introduction

Melatonin is a pleiotropic hormone produced by both prokaryotes and eukaryotes. One of the prominent functions of melatonin in humans is the regulation of our circadian rhythm. The synthesis and release of melatonin from the pineal gland are synchronized with the diurnal rhythm, and reach a peak at night time. Melatonin primarily signals through the activation of G protein-coupled receptors (GPCRs), and two subtypes of human melatonin receptors have been identified and denoted as type-1 (MT_1_) and type-2 (MT_2_) receptors [1,2]. Both MT_1_ and MT_2_ receptors belong to the Class A GPCR family, and they are expressed in a variety of tissues that range from multiple brain regions to numerous peripheral tissues such as the liver and skin [3]. The wide distribution of melatonin receptors implies that the therapeutic potentials of melatoninergic ligands can go far beyond the treatment of sleep disorders [4]. Indeed, melatonin receptors are implicated in the pathophysiology of Alzheimer’s disease [5,6], major depression [7], type 2 diabetes [8], and several other disorders [4]. Irrespective of the therapeutic areas, the beneficial effects of melatonin are invariably manifested through the melatonin receptors and their corresponding G protein-regulated signaling networks.

MT_1_ and MT_2_ receptors are commonly classified as G_i_-coupled receptors because their cellular signals are blocked upon the ADP-ribosylation of Gα_i/o_ proteins by pertussis toxin (PTX) [9]. Melatonin inhibits adenylyl cyclase (AC) through G_i_ proteins in many different types of cells and tissues [10,11,12,13,14] to exert its functions, including the regulation of the circadian rhythm and insulin secretion [15,16]. Paradoxically, and despite their demonstrated coupling to G_i_ proteins, the melatonin receptors are seemingly also able to stimulate the production of intracellular cAMP. Several studies have reported melatonin-induced cAMP formation in different cell types that express the MT_1_ or MT_2_ receptor endogenously or upon transfection [17,18,19,20,21]. The expression of the human MT_1_ receptor in the SH-SY5Y neuroblastoma cell line produced a different phenotypic outcome compared to its expression in other mammalian cell lines commonly used in heterologous expression studies. Instead of the inhibition of AC as observed in many cellular models [10,11,12,13,14,22], melatonin acutely stimulated cAMP synthesis and potentiated forskolin-stimulated cAMP production in SH-SY5Y cells [17]. This intriguing observation was presumed to arise from the indirect regulation of AC, as the melatonin-induced response in SH-SY5Y cells requires the mobilization of internal calcium stores [17], and because the protein kinase C (PKC)-mediated activation of distinct isoforms of AC (e.g., AC2 and AC4) has previously been demonstrated [23,24]. Although plausible, the proposed mechanism would entail the release of Gβγ dimer from G_i_ proteins that are activated by the melatonin receptors [25], but the melatonin-mediated stimulation of cAMP formation in SH-SY5Y cells is independent of functional G_i_ proteins [17].

As exemplified by the β-adrenergic receptors [26,27], it has long been known that GPCRs can dually couple to G_i_ and G_s_ proteins, and this possibility cannot be entirely excluded for the melatonin receptors. There is indeed evidence of G_s_ coupling to melatonin receptors. Melatonin has been shown to dose-dependently stimulate cAMP synthesis in rabbit nonpigmented ciliary epithelial cells [20] and chick adenohypophysis cells [21], while the MT_1_-mediated stimulation of cAMP in 22Rv1 and RWPE-1 human prostate epithelial cells is abolished upon the knockdown of Gα_s_, but not of Gα_i2_ [18]. The functional characterization of melatonin receptors from the Atlantic salmon in mammalian COS-7, CHO, and SH-SY5Y cells has also demonstrated a robust stimulation of a cAMP-driven CRE-Luc reporter through melatonin [28]. Moreover, the ability of melatonin to potentiate forskolin-induced cAMP production in different cell types [17,19,28,29] may reflect receptor coupling to G_s_, because all isoforms of AC respond synergistically to Gα_s_ and forskolin stimulations [30]. Given the contradictory reports on the regulation of intracellular cAMP levels by melatonin, it remains unresolved as to whether the melatonin receptors are dually coupled to G_i_ and G_s_ proteins.

Since G_s_-dependent signals appear to participate in multiple biological actions of melatonin, ranging from the regulation of extracellular signal-regulated kinase (ERK) [19,21], c-Jun N-terminal kinase (JNK) [24], Cl^−^ conductance [20,29], to lipolysis [31], it is pertinent to establish if MT_1_ and MT_2_ are indeed capable of directly stimulating G_s_. Given that classical (e.g., [^3^H]adenine, [^32^P]ATP, RIA, and ELISA) as well as contemporary (e.g., enzyme fragment complementation) methods for assaying intracellular cAMP levels often require one or more incubation steps and subsequent cell lysis, they may not be ideal for determining dual coupling to G_s_ and G_i_, which could be temporally regulated [27]. We therefore employed a highly sensitive luminescent biosensor (GloSensor^TM^) to monitor real-time changes in intracellular cAMP in intact live HEK293 cells expressing the MT_1_ or MT_2_ receptor. The GloSensor^TM^ assay utilizes a mutant form of *Photinus pyralis* luciferase that was engineered to produce light upon binding cAMP [32], and it has been successfully utilized to study the regulation of cAMP by different GPCRs in HEK293 cells [33,34]. Here, we report that the activation of MT_1_ (but not MT_2_) leads to a robust enhancement of forskolin-stimulated cAMP formation in a Gα_s_-dependent manner.

## 2. Results

### 2.1. Realtime Detection of cAMP in Live HEK293 Cells

In our experiments, the HEK293 cells were used for the heterologous expression of the MT_1_ or MT_2_ receptors, with the advantage of providing identical cellular background for the comparison of the signaling difference(s) between the two melatonin receptor subtypes. We first established the ability of the GloSensor^TM^ cAMP assay to detect changes in the intracellular cAMP concentration in live HEK293 cells upon stimulation by forskolin and isoproterenol. Forskolin is a natural diterpene that increases the cAMP level in cells by directly activating the ACs and is a common tool for studying the activities of G_i_ or G_s_ proteins [35]. Isoproterenol activates G_s_-coupled β_2_-adrenergic receptors (β_2_ARs) and is known to increase cAMP production in HEK293 cells by stimulating the endogenously expressed β_2_AR [36,37]. The biosensor pGloSensor^TM^-22F (0.5 μg) and MT_1_ (0.5 μg) were introduced into HEK293 cells on 60 mm dishes through PEI transfection. Transfectants were re-seeded onto 96-well plates one day later and then assayed after another day. As illustrated in Figure 1A (left panel), the application of forskolin at increasing concentrations (100 nM to 10 μM) rapidly increased the intracellular cAMP level, with a peak occurring at ~15 min. At the highest concentration of forskolin tested (i.e., 10 μM), the stimulated response was sustained over a period of 60 min (Figure 1A, left panel). Likewise, a concentration-dependent (10 pM to 10 μM) stimulation of cAMP formation by isoproterenol was clearly detected in these live cells (Figure 1B, left panel). The responses driven by isoproterenol were more rapid with peak attainment at ~5 min, and were transient rather than sustained (Figure 1B, left panel). The observed characteristics of isoproterenol are typical of GPCR-regulated pathways. These studies demonstrate the real-time detection of intracellular cAMP in live HEK293 cells.

### 2.2. Atypical Regulations of cAMP by MT_1_

We then examined the real-time cAMP response of the same transfectants upon a simultaneous challenge with a saturating concentration (100 nM) of melatonin. As a G_i_-coupled receptor, the activation of MT_1_ is expected to suppress the stimulatory actions of forskolin and isoproterenol. Surprisingly, the simultaneous addition of melatonin significantly stimulated the forskolin response across the different concentrations of forskolin tested, with up to a six-fold enhancement at 10 μM of forskolin (Figure 1A, right panel). In contrast, melatonin inhibited isoproterenol-stimulated cAMP formation in the MT_1_ transfectants (Figure 1B, right panel), as one would expect for a G_i_-coupled receptor agonist; melatonin inhibited the isoproterenol response by ~55%. The contrasting actions of melatonin on forskolin- and isoproterenol-stimulated cAMP formation are clearly illustrated upon plotting the cAMP-driven luminescence against the concentrations of forskolin/isoproterenol used in the absence or presence of 100 nM melatonin (Figure 1C).

When similar studies were performed on HEK293 cells co-expressing MT_2_ and the biosensor pGloSensor^TM^-22F, both forskolin and isoproterenol stimulated real-time productions of cAMP in concentration-dependent manners (Figure 2A,B, left panels). Melatonin, however, neither enhanced nor inhibited the forskolin-stimulated response in the MT_2_ transfectants (Figure 2A, right panel; Figure 2C, left panel), but remained capable of suppressing isoproterenol-induced cAMP formation (Figure 2B, right panel). The magnitude of the melatonin-mediated inhibition of the isoproterenol response was weaker in the MT_2_ transfectants compared to that in MT_1_ transfectants (Figure 2C, right panel).

Next, we investigated the effect of different concentrations of melatonin on the cAMP stimulation by 10 μM forskolin or 1 μM isoproterenol. The cAMP response was measured at 15 min or 5 min after forskolin or isoproterenol treatment, respectively. Interestingly, in the absence of agonists, the overexpression of either MT_1_ or MT_2_ receptors significantly suppressed the cAMP stimulation by both forskolin and isoproterenol compared to the pcDNA3.1 empty vector control, with a ~60% reduction in the basal cAMP level (Figure 3A). In contrast, no inhibition was observed in transfectants overexpressing the G_i_-coupled dopamine D_2_ receptor (D_2_R; Figure 3A). This result suggests the existence of a constitutive inhibitory activity of the melatonin receptors, which was absent from D_2_R.

The ligand-induced receptor activity on cAMP production was then investigated, and corresponding dose response curves were generated. Increasing concentrations of melatonin or quinpirole (agonist of D_2_R) were co-administered with 10 μM forskolin or 1 μM isoproterenol in the MT_1_-, MT_2_-, or D_2_R-expressing cells. In the MT_1_ transfectants_,_ melatonin potentiated the forskolin-stimulated cAMP response across various melatonin concentrations tested (1 nM to 1 μM) as compared to cells treated with forskolin alone (Figure 3B, upper left panel). However, no significant change in the forskolin-stimulated cAMP level was observed in the MT_2_ transfectants treated with different concentrations of melatonin (Figure 3B, upper left panel). In contrast to the forskolin-induced cAMP response, the application of melatonin significantly inhibited isoproterenol-induced cAMP production in both MT_1_ and MT_2_ transfectants in a dose-dependent manner, with a maximal reduction of ~50% and ~40% in cAMP level, respectively (Figure 3B, lower left panel). On the other hand, as expected for a typical G_i_-coupled receptor, the activation of D_2_R by quinpirole dose-dependently suppressed both forskolin- and isoproterenol-induced cAMP production, with the maximal of a more than 80% reduction in both forskolin- and isoproterenol-stimulated responses (Figure 3B, upper and lower right panels). These results demonstrate an atypical stimulatory action of melatonin on the forskolin-induced cAMP response by the MT_1_ receptor.

### 2.3. cAMP Stimulation by MT_1_ Is Independent of G_i_ Signaling

To dissect the signaling mechanism underlying the atypical stimulation on the forskolin-induced cAMP response by the MT_1_ receptor, we first assessed the involvement of G_i_ signaling since the Gβγ dimer from G_i_ proteins has been shown to contribute to the activation of several AC isoforms [38]. We examined the effect of PTX, a toxin widely used to block the G_i_ signaling pathway [39], on the cAMP response in HEK293 cells expressing MT_1_, MT_2_, or D_2_R. As illustrated in Figure 4A, the overnight treatment of cells with PTX (100 ng/mL) failed to suppress or block the promoting effect of 100 nM melatonin on forskolin-induced cAMP formation in the MT_1_ transfectants, whereas the same PTX treatment completely abolished the inhibitory effect of quinpirole on forskolin-induced cAMP production in cells expressing D_2_R (Figure 4B). This result excludes the involvement of G_i_ signaling in the MT_1_-mediated potentiation of cAMP stimulation by forskolin.

### 2.4. MT_1_ Stimulates cAMP Production through Gα_s_ Protein

G_s_ protein has been reported to potentiate the response of AC (isoforms type II, V, and VI) toward forskolin [40]. As illustrated in Figure 5A, the activation of endogenous G_s_-coupled β_2_AR indeed resulted in the potentiation of the forskolin response in HEK293 cells. Likewise, the ability of melatonin to significantly enhance forskolin-induced cAMP production in MT_1_ transfectants (Figure 1A, right panel; Figure 3B, upper left panel) could also be mediated through G_s_. However, unlike the activation of β_2_AR by isoproterenol, melatonin alone did not stimulate cAMP formation in MT_1_ transfectants (Figure 5A). This might be attributed to a stronger coupling of the MT_1_ receptor to G_i_ than to G_s_ proteins, resulting in constitutive G_i_ activity and a lower forskolin response, as observed in Figure 3A. If so, the increased expression of G_s_ may shift the coupling specificity of MT_1_ away from G_i_ and enable melatonin to stimulate cAMP production in the absence of forskolin. The overexpression of Gα_s_ sensitized the forskolin-induced cAMP production (Figure 5A,B) and resulted in an apparent impairment of the ability of isoproterenol to further enhance the forskolin response. Similar results were obtained with MT_1_. The simultaneous addition of melatonin significantly enhanced the forskolin-induced cAMP response in the MT_1_/Gα_s_ transfectants (Figure 5B), albeit to a lesser extent as compared to the MT_1_ transfection alone (Figure 5A). More importantly, melatonin alone significantly stimulated cAMP formation in MT_1_/Gα_s_ transfectants (Figure 5B). The co-expression of Gα_s_ with MT_2_ or D_2_R, however, did not enable the corresponding agonists to trigger cAMP production (Figure 5B), thereby demonstrating a unique G_s_-coupling property of the MT_1_ receptor.

### 2.5. Molecular Modeling Predicts a Viable MT_1_-G_s_ Complex

Given the prior demonstration of the dual coupling of the β_2_AR to both G_s_ and G_i_ proteins [26,41] and numerous hints of MT_1_ being capable of stimulating cAMP production [17,18,19,20], it seems plausible that MT_1_ can interact with G_s_. We used computational tools to generate a MT_1_-G_s_ structure and compared the stability of this complex against the resolved structure of MT_1_-G_i_ [PDB code 7VGZ; [42]]. An MT_1_-G_s_ complex was constructed using Modeller10.4 and the top 10 models out of 1,000 ensembles generated via LD (see Section 4) were evaluated at their receptor-G protein interface. The top model for MT_1_-G_s_ is illustrated in Figure 6A; it closely resembles the overall topological configuration of MT_1_-G_i_ (also shown in Figure 6A). This similarity in the arrangement of G protein isoforms potentially explains the comparable energy functions observed. The MT_1_-G_s_ complex has a low dissociation constant, indicating high-affinity binding, and a free energy of binding that is comparable to that of the MT_1_-G_i_ complex (as depicted in Figure 6B). These findings provide further evidence supporting the potential feasibility of coupling G_s_ proteins to MT_1_. A viable MT_1_-G_s_ complex should exhibit a low dissociation constant for high-affinity binding and a favorable free energy of binding. The measured free energy of approximately −10 kcal/mol and a dissociation constant of 1 × 10^−5^ mM (Figure 6B) are consistent with these requirements. Additionally, the interface area of approximately 1000 $\overset{{}^{\circ}}{A}$, as observed in the MT_1_-G_s_ complex, is comparable to that of the MT_1_-G_i_ complex (Figure 6B). These measurements provide additional support for the idea that MT_1_ can form signaling assemblies with both G_i_ and G_s_ proteins through dual coupling. The observed similarities in stability during the formation of these signaling assemblies indicate that MT_1_ can effectively interact with and activate both G_i_ and G_s_ proteins, contributing to the diversity and complexity of its signaling capabilities.

However, a significant difference was observed in the dissociation constant, with MT_1_-G_i_ displaying a value of 1 × 10^−4^ mM and MT_1_-G_s_ having a 10-fold higher affinity at 1 × 10^−5^ mM (Figure 6B). Although this suggests that the MT_1_-G_s_ complex is more stable than the MT_1_-G_i_ complex, the probability or preference of G protein coupling with MT_1_ cannot be solely concluded based on their dissociation constants, because other factors such as the relative abundance of individual protein would affect the equilibrium. Collectively, our predicted model demonstrates that following activation, the orientations and energy functions favor the coupling of G_s_ proteins in a manner that closely resembles the coupling of G_i_ proteins.

## 3. Discussion

The long-standing conundrum of the G_i_-coupled melatonin receptors stimulating the formation of cAMP in specific cell types has raised an important question of whether such observations are due to the direct or indirect regulation of AC. The ability of melatonin receptors to activate G_q_ [9,43] provides an avenue for melatonin to stimulate specific isoforms of AC via PKC-mediated phosphorylation [23]. A more tantalizing explanation is that the melatonin receptors can dually couple to G_s_ and G_i_ to directly regulate AC in a manner similar to that of the well-established β_2_AR [41]. Direct and indirect activations of effectors can typically be discerned by their temporal characteristics. The direct stimulation of AC by G_s_ should be detectable immediately, while regulations through phosphorylation events would be delayed. Hence, we used the GloSensor^TM^ system to examine the real-time cAMP responses driven by the melatonin receptors in HEK293 cells. Our data revealed an atypical cAMP regulation by the MT_1_ receptor. The results from both the functional and molecular modeling experiments support the notion that MT_1_ can interact productively with G_s_ to stimulate AC, and the magnitude of this response is amplified upon the sensitization/activation of AC by forskolin. Given the transient nature of G_s_-induced AC activity (Figure 1B and Figure 2B), MT_1_/G_s_-mediated cAMP formation might be masked in standard assays that require extended manipulations or incubations such as radioimmunoassays and reporter gene assays.

It has long been considered that both MT_1_ and MT_2_ receptors are primarily coupled to G_i_ proteins and inhibit the cAMP level [44]. Indeed, the current study observed a constitutive suppressive effect of MT_1_ and MT_2_ on the forskolin- and isoproterenol-induced cAMP production (Figure 3A). The application of melatonin significantly inhibited the isoproterenol-induced cAMP response in both MT_1_- and MT_2_-expressing cells, as one would expect for G_i_-coupled receptors. These data support the canonical coupling of G_i_ protein to the MT_1_ and MT_2_ receptors. However, contradictory to what was observed for the isoproterenol-stimulated cAMP responses, the application of melatonin led to a prominent potentiation of forskolin-induced cAMP production in MT_1_-expressing cells whereas no effect was observed in the MT_2_-expressing cells. The contrasting actions of melatonin on forskolin- and isoproterenol-stimulated cAMP formation in the MT_1_-expressing cells imply the existence of an atypical and alternative signaling mechanism mediated by the MT_1_ receptor. The results from the G_s_ co-expression experiment suggest that MT_1_ indeed possesses the ability to directly activate G_s_, since the increased availability of G_s_ allowed for the lucid detection of the MT_1_-mediated stimulation of AC (Figure 5), presumably by favoring G_s_ to compete against G_i_ in binding to MT_1_. Although such a response may be considered as “forced coupling”, neither MT_2_ nor D_2_R was capable of elevating cAMP levels in the Gα_s_ transfectants (Figure 5). The favorable parameters of the predicted MT_1_-G_s_ complex generated by molecular modeling (Figure 6) suggest the possible existence of this complex alongside the known MT_1_-G_i_ structure [42]. The predicted affinity of MT_1_ for G_s_ is comparable to, if not higher than, that for G_i_.

Our results from the real-time GloSensor^TM^ cAMP assay and molecular modeling tend to corroborate with previous reports on the stimulatory effect of melatonin on cAMP production [17,18,19,20,21] and support the peculiar capacity of MT_1_ to signal through G_s_ [19,24]. In the same cell system, the activation of the prototypical G_i_-coupled receptor, D_2_R, significantly suppressed the forskolin-induced cAMP synthesis. Though the Gβγ dimer from G_i_ proteins has been shown to contribute to the activation of several AC isoforms [38], our data from the PTX experiment excluded the involvement of G_i_ signaling in the MT_1_-mediated potentiation of cAMP stimulation by forskolin. Furthermore, the co-expression experiments indeed suggest a functional coupling of G_s_ to the MT_1_ receptor. Firstly, with the overexpression of G_s_ protein, melatonin could initiate a significant rise in the intracellular cAMP level through MT_1_ even in the absence of forskolin, whereas no detectable changes were observed for the MT_2_ or D_2_R transfectants. Paradoxically, as opposed to what was observed for the forskolin-induced response, the activation of MT_1_ significantly suppressed the isoproterenol-stimulated cAMP formation, which implicates the functional coupling of G_i_ protein to the receptor. Hence, the present study explicitly demonstrates the dual activation of G_s_ and G_i_ signaling pathways by the MT_1_ receptor in the HEK293 cell system. The various scenarios under which MT_1_ may stimulate or inhibit the activity of AC are depicted in Figure 7.

Dual coupling has been observed for multiple GPCRs, including the prototypical β_2_AR, which can signal through both G_s_ and G_i_ proteins [26,27]. Based on the contrasting actions of melatonin on forskolin- and isoproterenol-stimulated cAMP formation in the MT_1_-expressing cells, we proposed that MT_1_ can bi-directionally regulate AC by alternating its signaling pathways between G_s_ and G_i_, which depends on the relative abundance of the G proteins in the cellular system (Figure 7). Hence, in the MT_1_ transfectants, the binding of melatonin activates MT_1_, resulting in the potentiation of a forskolin-stimulated cAMP response owing to its strong synergism with activated G_s_ proteins. It should be noted that the activation of the endogenous β_2_AR receptor by isoproterenol can markedly stimulate cAMP level in HEK293 cells, whereas cAMP stimulation through MT_1_ activation necessities the overexpression of G_s_ protein, which implies that the G_s_ coupling affinity is relatively weaker for the MT_1_ receptor. It is therefore not surprising that with the simultaneous addition of melatonin and isoproterenol, the β_2_AR outcompetes MT_1_ for the endogenous G_s_ protein. As a result, MT_1_ signaling switches to the G_i_ pathway due to its coupling to the spatially more abundant G_i_ protein. Therefore, unlike the forskolin-induced cAMP response, melatonin significantly inhibited isoproterenol-induced cAMP synthesis in the MT_1_-expressing cells. This proposed mechanism enables the MT_1_ receptor to signal through alternative G protein pathways in the same cellular context.

The differential signaling pathways mediated by the MT_1_ receptor under different stimulations (forskolin or β_2_AR receptor) could be observed in a specific cellular context. An illustrative example of cell context-dependent signaling is the G_i_-mediated stimulation of type II adenylyl cyclase (ACII) through the βγ dimer. For biological systems with the expression of ACII, the activation of G_i_-coupled receptors could lead to an upregulation in the intracellular cAMP level in the presence of activated G_s_ or PKC [23]. In addition, Gα_s_ was shown to sensitize forskolin-induced cAMP production in a cell-context dependent manner [40,45]. With the recently resolved cryo-EM structures of adenylyl cyclase 9 (AC9) [46], the distinct sites for interaction with Gα_s_ and forskolin on AC9 were identified. Gα_s_ was shown to interact with the C2a catalytic domain of AC9, whereas forskolin was shown to bind to an allosteric site adjacent to the catalytic site. These data revealed the structural basis of the commonly known synergistic activation of AC by Gα_s_ and forskolin. Similarly, the MT_1_ receptor/G protein coupling efficiency is highly cell context-dependent, which could be modulated by the relative abundance of the G proteins in the cellular system; G_s_ overexpression in the HEK293 cell system increased the proportion of MT_1_ receptor coupling to the G_s_ proteins, thereby leading to a lucid detection of an elevated cAMP level upon receptor activation. Therefore, MT_1_-mediated cAMP stimulation via coupling to G_s_ was observed only under certain circumstances, depending on the relative abundance of the G proteins. An illustrative example for the coupling of MT_1_ receptor to the G_s_ protein is the human prostate epithelial cells, which express high levels of MT_1_ and MT_2_ receptors as well as G_s_ protein [18]. As demonstrated in the study using human prostate cancer cell line 22Rv1 and human prostate epithelial cell line RWPE-1, the application of 2-iodomelatonin increased the intracellular level of cAMP, and this effect was blocked by the nonselective MT_1_/MT_2_ receptor antagonist luzindole but not the selective MT_2_ receptor antagonist, 4P-PDOT. In this study, the knockdown of G_s_ abrogated the stimulatory effect of 2-iodomelatonin on cAMP level, thereby demonstrating the stimulatory effect of endogenous MT_1_ signaling on cAMP level by coupling to the G_s_ protein. Hence, it would be rationale to postulate that for cells with high expression levels of both MT_1_ and G_s_ protein, the activation of MT_1_ may upregulate cAMP level through the activation of AC.

Interestingly, the present study also revealed a constitutive inhibitory effect of MT_1_ and MT_2_ on the cAMP level. Indeed, most GPCRs do exhibit some degree of basal activity. With regard to melatonin receptors, a previous study has suggested that the MT_1_ receptor is tightly pre-coupled to G_i_ and is spontaneously active [47]. The affinity of G_i_ toward MT_1_ was not altered in the presence of melatonin, and the GTPγS binding assay revealed a constitutive activity of G_i_ in the reconstituted membrane of MT_1_ and G_i1_ [47]. In addition, MT_2_ has been reported to activate Gα_i1_ and Gα_z_ proteins spontaneously based on BRET assays [48].

The difference between the present study and previous publications is possibly due to the alternative detection methods. Most of the published studies detected cAMP pathway through radioimmunoassays or radiometric assays [44,49,50], which require a prolonged drug incubation time (agonist and forskolin) [24] or the treatment of cells with agonist prior to the addition of forskolin [51]. The change in cAMP level was often determined as an accumulative result over time instead of reflecting the real-time level, which could be affected by downstream effectors through indirect/feedback mechanisms. Using the [^3^H]cAMP accumulation assay as an example, the cells labeled with [^3^H]adenine would be incubated with drugs for 30 min at 37 °C before the collection of the total labeled nucleotide fraction from the cell lysates, and the [^3^H]cAMP level would be determined using the ratio of the [^3^H]cAMP fraction over total nucleotide [24]. However, as detected through the real-time GloSensor^TM^ cAMP assay, the activity of forskolin or isoproterenol on cAMP production started to diminish after 15 min or 5 min of drug treatment, respectively. Hence, the GloSensor^TM^ cAMP assay would be a better means to reflect the real-time changes in cAMP levels in live intact cell systems.

In conclusion, the present study demonstrates the functional coupling of G_s_ protein to the melatonin MT_1_, but not the MT_2_ receptor. The stimulation of MT_1_ can lead to the dual activation of the G_s_ and G_i_ pathways in the HEK293 cells. This ability of MT_1_ to regulate AC bi-directionally may explain the seemingly contradictory effects of melatonin on intracellular cAMP levels observed in different cell types, since the net production of cAMP will depend on the relative abundance of G_s_, G_i_, MT_1_, and MT_2_ and on the efficiency of the respective receptor/G protein coupling. The further characterization of the signaling property of the MT_1_ receptor will be required to better elucidate its functional significance in various cellular systems. In addition, the GloSensor^TM^ cAMP assay may be a better option for future studies on GPCR-mediated cAMP regulation as it enables the real-time determination of the highly dynamic intracellular cAMP level.

## 4. Materials and Methods

### 4.1. Materials

The cDNAs encoding MT_1_ and MT_2_ were kindly provided by Dr. Steven Reppert (Massachusetts General Hospital, Boston, MA, USA). The D_2_R (NM_000795) and Gα_s_ (NM_000516) cDNA were purchased from the cDNA Resource Center (www.cdna.org; accessed on 25 January 2024).

### 4.2. Cell Culture and Transfection

Human Embryonic Kidney 293 cells (HEK293 cells; ATCC, CRL-1573) were cultured in Minimum Essential Medium (MEM; Gibco, Thermo Fisher Scientific, Waltham, MA, USA) supplemented with 10% (*v*/*v*) heat-inactivated fetal bovine serum (Gibco), 100 µg/mL streptomycin and 100 units/mL penicillin, and grown in a humidified 37 °C, 5% CO_2_ incubator. The day before transfection, the cells were seeded onto 60 mm dishes at a density of 3.5 × 10^5^ cells/mL in 4 mL MEM and incubated at 37 °C with 5% CO_2_ overnight. Transient transfection was performed using Polyethylenimine 25 kDa linear (PEI) transfection reagent (Polysciences, Inc., Warrington, PA, USA) in a ratio of 1:4 (DNA in µg/PEI in µL). The empty vector pcDNA3.1 was used to balance the amount of DNA in each transfection.

### 4.3. GloSensor cAMP Assay

A GloSensor cAMP assay was performed according to the manufacturer’s protocol (Promega, Madison, WI, USA). Briefly, HEK293 cells were transiently transfected with the pGloSensor-22F cAMP plasmid and the desired receptor or G protein constructs in a 60 mm dish. After 24 h, transfected cells were re-plated onto a white 96-well cell culture plate and incubated at 37 °C with 5% CO_2_ overnight. The growth medium was then replaced with equilibration medium (HBSS with 10 mM HEPES, pH 7.4) containing the GloSensor cAMP Reagent. The cells were incubated with the GloSensor cAMP Reagent at room temperature for 20 min, and the luminescence was monitored to obtain a steady-state basal signal. To study the cAMP response in HEK293 cells expressing MT_1_ or MT_2_, transfected cells were treated with increasing concentrations of forskolin or isoproterenol in the absence or presence of 100 nM melatonin. To study the receptor-mediated cAMP response in HEK293 cells expressing MT_1_, MT_2_, or D_2_R, 10 µM forskolin or 1 μM isoproterenol along with varying concentrations of melatonin or quinpirole were added to the cells. The effect of PTX on cAMP regulation by GPCRs was examined in transfected cells pretreated overnight with vehicle or PTX (100 ng/mL) prior to 10 μM forskolin stimulation in the absence or presence of 100 nM melatonin or quinpirole. Luminescence was measured using the SpectraMax L microplate reader (Molecular Devices, LLC., San Jose, CA, USA). The GraphPad Prism 9 software was used for graph plotting, and concentration-response curves were generated using one-site competition non-linear regression.

### 4.4. Loop Modeling and the Generation of Initial Structures

The structure of the intracellular loop 3 (IL3) in GPCRs is often unresolved in crystal structures due to its inherent flexibility, despite its crucial role in G protein coupling [52]. To overcome this challenge, we used Modeller10.4 to reconstruct the complete active MT_1_ receptor [PDB code 7VGZ; [42]] and generated a specific active conformation for IL3 in the MT_1_ receptor. The missing IL3 segment was identified using EMBOSS Needle, a pairwise alignment tool that compares a protein sequence from the PDB structure with that in the FASTA file [53]. The final model of the refined loop was selected based on the DOPE scoring function and subsequently validated using PROCHECK (https://www.ebi.ac.uk/thornton-srv/software/PROCHECK/; accessed on 25 January 2024) [54,55]. To establish reasonable starting conformations of the MT_1_-G_s_ complex, the generated MT_1_ receptor structure and the G_s_ protein [PDB code 8DCS; [56]] were superimposed onto the crystal structures of the active MT_1_-G_i_ complex [PDB code 7VGZ; [42]] and the active β_1_-adrenergic receptor [PDB code 8DCS; [56]]. These reference templates were chosen because they share a similar orientation and conformation during G protein coupling. It is important to note that since no crystallized structure of the MT_1_-G_s_ complex is currently available, and different orientations may be required for receptor-G protein interactions, we cannot disregard the possibility of various modes of interactions. Hence, two different reference template structures were employed to impose the MT_1_-G_s_ complex. These positions were chosen to ensure compatibility with the binding pockets of the receptor and G protein [57,58].

### 4.5. Determining Receptor-G Protein Interface via Local Docking

Due to limited knowledge and the dynamic nature of interactions between the receptor and G protein complex [59], Rosetta Local Dock (LD) was employed to explore potential modes of G_s_ protein coupling with MT_1_. We focused on investigating the interface between MT_1_ and the heterotrimeric G_s_ by identifying and comparing probable regions of interaction. The superimposed MT_1_-G_s_ complex was then prepared for LD by removing extraneous information, such as water molecules that are not recognized by the Rosetta program. LD was performed by repacking the side chains, optimizing the initial structure of the predicted model, and generating 1000 ensembles. Default docking parameters, such as a docking partner distance of 3 $\overset{{}^{\circ}}{A}$ and rotation of 8°, were employed [57]. To prevent unintended perturbations between multi-chain proteins, the α-subunit and βγ-dimer of the G_s_ protein were restricted by optimizing them together during the entire docking process [58]. These ensembles were sorted based on their lowest interface scores and total Rosetta scores, and the top 10 models were selected for further analysis.

### 4.6. Scoring and Analysis of the Predicted Complex Models

The top 10 predicted models selected based on their lowest Rosetta interface scores [60] were submitted to the PRODIGY web server (bianca.science.uu.nl/prodigy/; accessed on 25 January 2024) for further analysis. The PRODIGY web server was used to evaluate the dissociation constant at 37 °C and the free energy of binding [61]. The interface areas between the MT_1_ and G_s_ proteins were measured using UCSF Chimera in the default mode [62]. For the visualization of the entire assemblies, the PyMOL molecular Graphics System 2.1 (Schrödiner, LLC., New York, NY, USA) was employed.

## Figures and Tables

**Figure 1 ijms-25-02919-f001:**
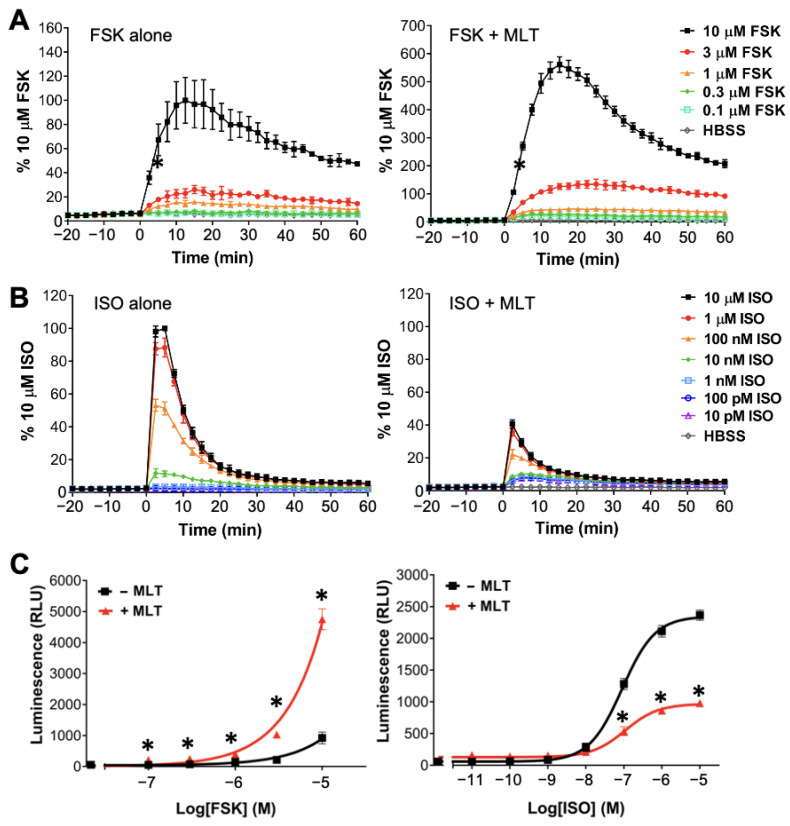
Kinetic profiles of the GloSensor^TM^ cAMP assay detecting the cAMP response in HEK293 cells transfected with 0.5 μg MT_1_ and 0.5 μg biosensor pGloSensor^TM^-22F. Transfected cells were treated with increasing concentrations of (**A**) forskolin (FSK) or (**B**) isoproterenol (ISO) in the absence (**left**) or presence (**right**) of 100 nM melatonin (MLT). Representative graphs of three independent experiments performed in triplicate are shown. Data are expressed as a percentage of the peak luminescence obtained under the 10 μM FSK or 10 μM ISO condition and shown as mean ± SD of the triplicate determinations. (**C**) Luminescence measured at 12.5 min for FSK treatment (**left**) or 2.5 min for the ISO treatment (**right**) after the addition of compounds were plotted against the concentrations of the forskolin/isoproterenol used in the absence or presence of 100 nM melatonin. Representative graphs of three independent experiments performed in triplicate are shown, and the data represent the mean ± SD of the triplicate determinations. A statistical analysis was performed using a two-tailed *t*-test; * *p* < 0.01.

**Figure 2 ijms-25-02919-f002:**
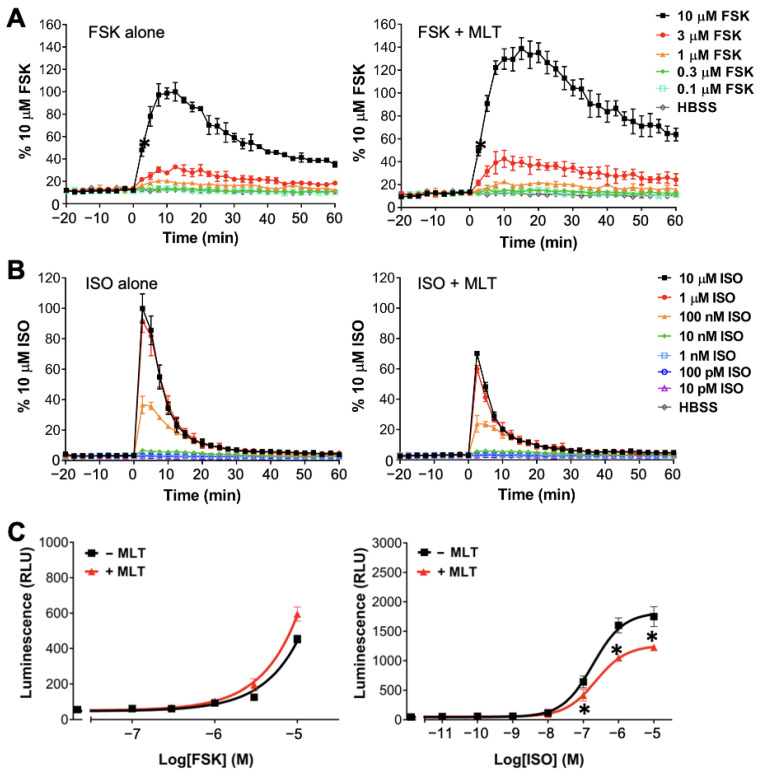
Kinetic profiles of the GloSensor^TM^ cAMP assay detecting cAMP response in HEK293 cells transfected with 0.5 μg MT_2_ and 0.5 μg biosensor pGloSensor^TM^-22F. (**A**–**C**) MT_2_ transfectants were treated as described in Figure 1, and the results were analyzed in the same manner. Representative graphs of three independent experiments performed in triplicate are shown, and the data represent the mean ± SD of the triplicate determinations. A statistical analysis was performed using a two-tailed *t*-test; * *p* < 0.01.

**Figure 3 ijms-25-02919-f003:**
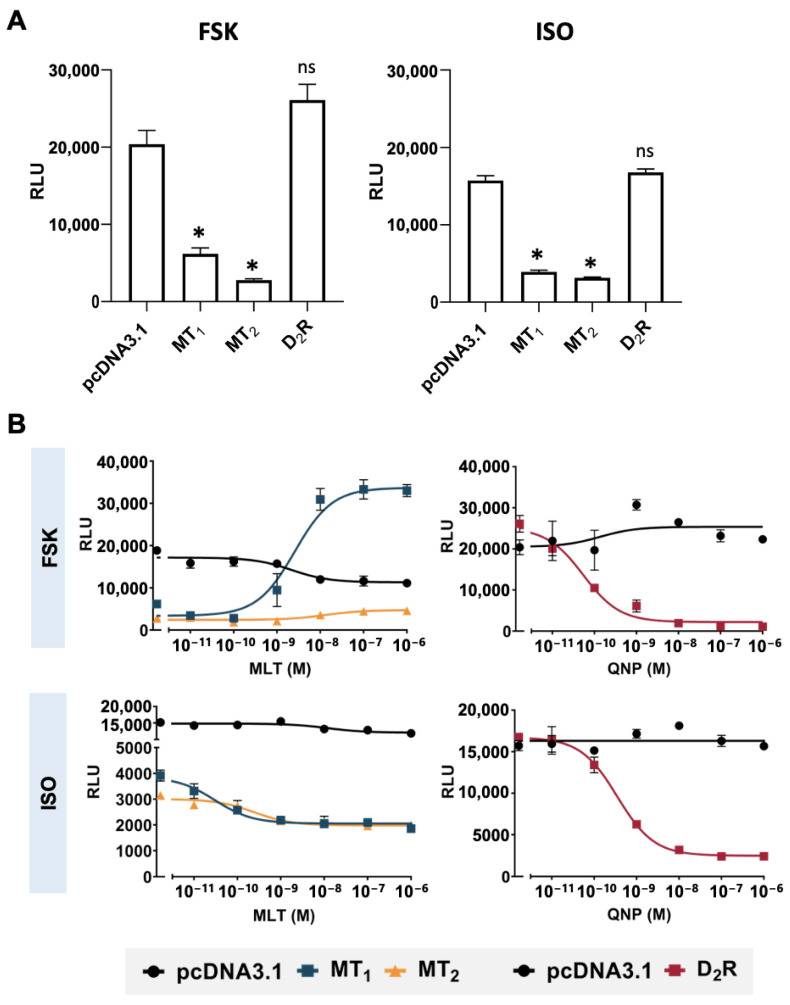
GloSensor^TM^ cAMP assay detecting the cAMP regulation in HEK 293 cells. Cells were transiently transfected with pcDNA3.1 empty vector control, MT_1_ (0.5 μg), MT_2_ (0.5 μg)_,_ or D_2_R as a G_i_-coupled receptor control (0.5 μg), together with the biosensor pGloSensor^TM^-22F (0.5 μg). (**A**) The cAMP levels measured at 15 min or 5 min for FSK (**left**) or ISO (**right**) treatment, respectively. (**B**) Effects of MLT or quinpirole (QNP) on FSK-(**upper**) or ISO-induced (**lower**) cAMP production in MT_1_ and MT_2_ (**left**), or D_2_R-transfected (**right**) HEK 293 cells. The data represent the mean ± SEM of the three independent experiments performed in triplicate. A statistical analysis was performed using ANOVA and Dunnett’s test; ns, not significant; * *p* < 0.01.

**Figure 4 ijms-25-02919-f004:**
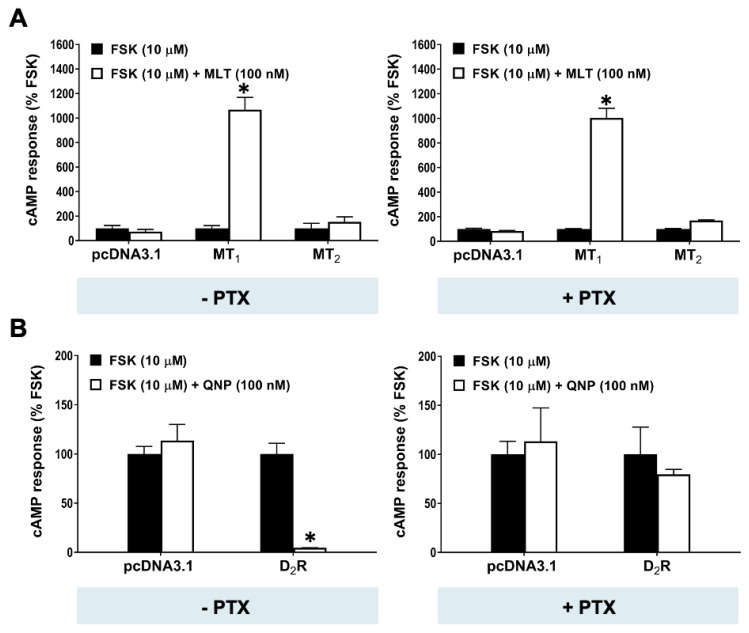
Effect of PTX treatment on cAMP response in HEK293 cells expressing melatonin MT_1_, MT_2_, or dopamine D_2_ receptors. HEK293 cells were transfected with pcDNA3.1 empty vector, MT_1_ (0.5 μg), MT_2_ (0.5 μg), or D_2_R (0.5 μg), together with the biosensor pGloSensor^TM^-22F (0.5 μg). Transfected HEK293 cells were subjected to an overnight vehicle or PTX (100 ng/mL) treatment prior to 10 μM FSK stimulation in the absence or presence of (**A**) 100 nM MLT or (**B**) 100 nM QNP. Representative graphs of three independent experiments performed in triplicate are shown. cAMP responses at 15 min are normalized to the response with forskolin alone and shown as the mean ± SD of the triplicate determinations. A statistical analysis was performed using a two-tailed *t*-test; * *p* < 0.01.

**Figure 5 ijms-25-02919-f005:**
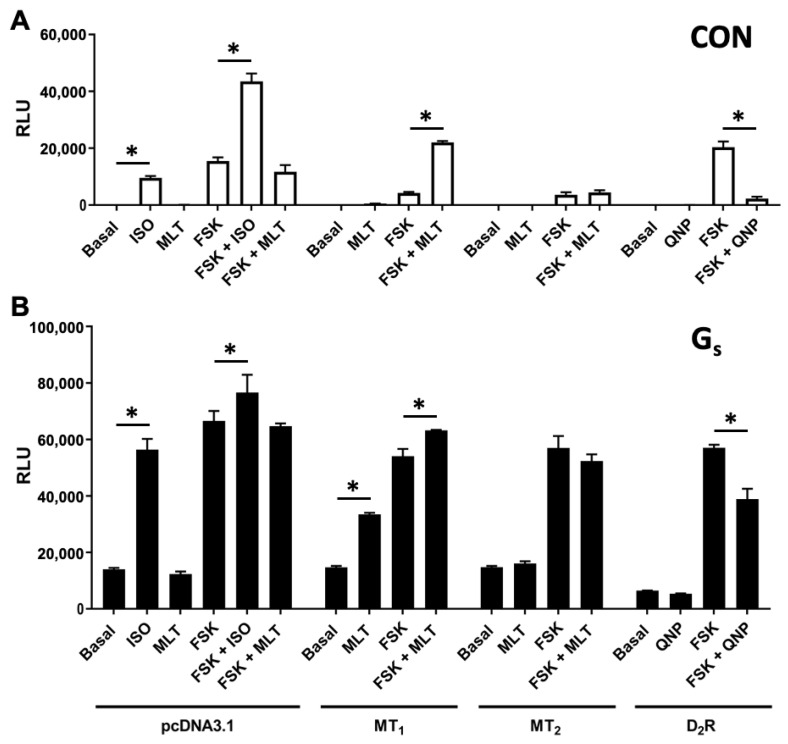
The cAMP regulation by GPCRs under different G protein systems. pcDNA3.1, MT_1_ (0.5 μg), MT_2_ (0.5 μg), and D_2_R (0.5 μg) were transiently transfected into HEK 293 cells together with pGloSensor^TM^-22F (0.5 μg), with or without the co-expression of G_s_ protein (0.5 μg). The cAMP regulation in HEK 293 cells (**A**) under an endogenous G protein system or (**B**) with the overexpression of G_s_ protein was detected using the GloSensor^TM^ cAMP assay. Representative graphs of three independent experiments performed in triplicate are shown. The RLU reading of the assay was plotted after a 15 min treatment with a vehicle (basal), ISO (100 nM), MLT (100 nM), or QNP (100 nM) with or without FSK (10 μM) as indicated. Data are shown as the mean ± SD of the triplicate determinations. A statistical analysis was performed using ANOVA and Dunnett’s test; * *p* < 0.01.

**Figure 6 ijms-25-02919-f006:**
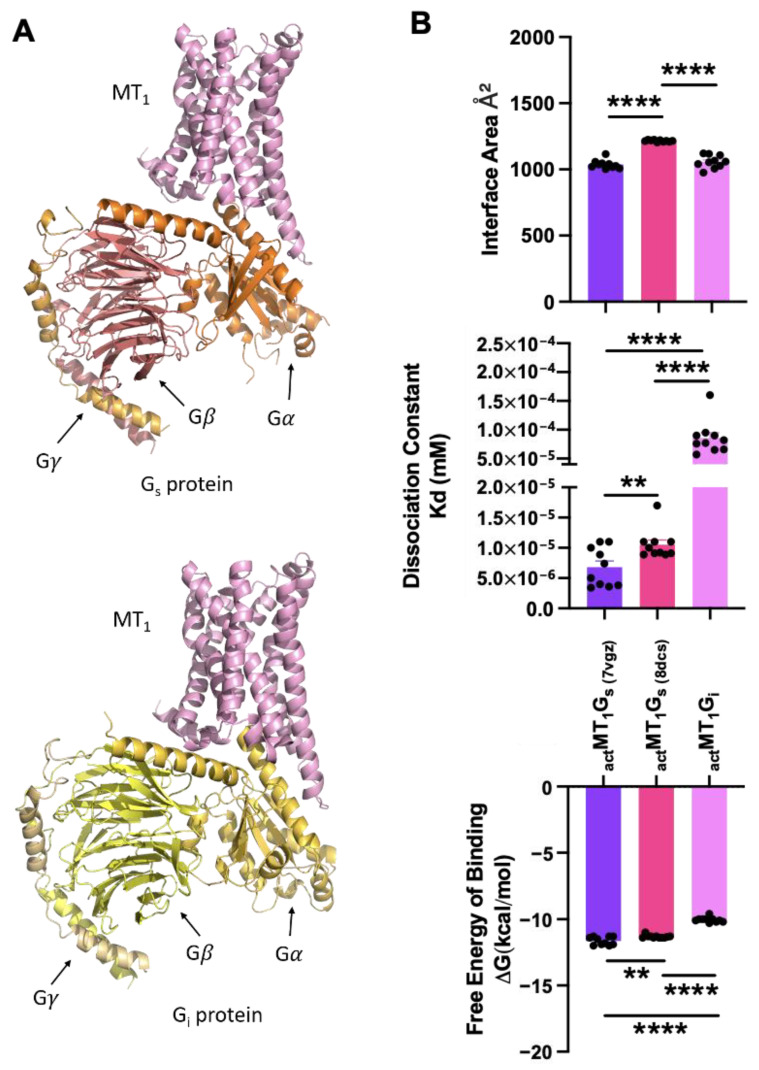
(**A**) Molecular models of the potential G_s_ protein (top) and G_i_ protein (bottom) coupling to the monomeric form of MT_1_. MT_1_ is depicted in pink; G_s_ protein subunits, in shades of orange; and G_i_ protein subunits, in shades of yellow. (**B**) Scoring parameters (interface area, dissociation constant, and free energy of binding) of the receptor-G protein interface. The values were obtained by averaging the scores of the top 10 predicted models for each receptor-G protein complex. _act_MT_1_G_s_ structures predicted using MT_1_G_i_ (PDB code 7VGZ) and β_1_AR-G_s_ (PDB code 8DCS) templates are labeled as _act_MT_1_G_s_ (7vgz) and _act_MT_1_G_s_ (8dcs), respectively. Mean ± SEM was determined. A two-tailed *t*-test was conducted for each complex; and significance levels are denoted as *** p* < 0.05, ***** p* < 0.01.

**Figure 7 ijms-25-02919-f007:**
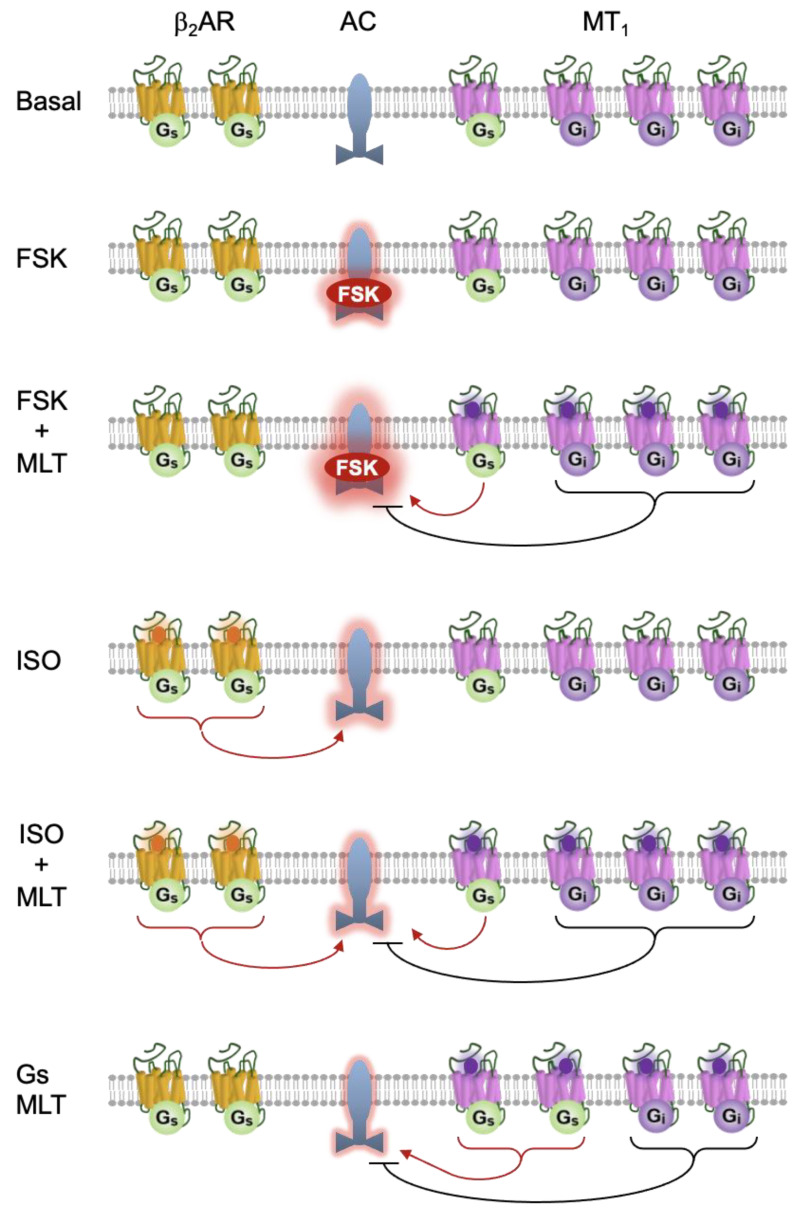
Schematic representation depicting the regulation of AC (blue) by the MT_1_ receptor (pink) under different conditions. In the basal state, MT_1_ is primarily coupled to G_i_ (purple) rather than to G_s_ (green). Forskolin (FSK) directly activates AC as indicated by a red halo. Although the activation of MT_1_ by melatonin (MLT; purple dot) tends to inhibit AC (black blunted arrow), the MT_1_/G_s_ signal (red line with arrow) synergizes with FSK to enhance AC activity (larger halo). The activation of the β_2_AR by isoproterenol (ISO; orange dot) also stimulates AC, which can be partially suppressed upon the activation of MT_1_. When the G_s_ expression is increased, the proportion of MT_1_ bound to G_s_ relative to G_i_ will increase, thereby allowing MLT to stimulate AC without the need of FSK.

## Data Availability

The data presented in this study are available on reasonable request from the corresponding author.

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
