# Peer review of "Real-Time Determination of Intracellular cAMP Reveals Functional Coupling of Gs Protein to the Melatonin MT1 Receptor"

_ijms, 2024, doi:10.3390/ijms25052919_

Round 1

Reviewer 1 Report

Comments and Suggestions for Authors

The authors nicely demonstrated the dual specificity of the human melatonin MT1 receptor. In a HEK293 cell system, they generated functional data (real time cAMP monitoring) of Gs protein coupling to the MT1 receptor, a usually Gi coupled receptor.

Although the presented data give a complete picture, there are some limitations that must be addressed:

First, the effects are only evident in an artificial cell culture system (transiently transfected HEK293 cells). Therefore, neither the expression levels of the receptors nor their ratios to the G-proteins and other GPCRs are known. Some Western blots would be helpful to show the expression levels of the proteins involved and compare them with the expression in relevant tissues.

Second, the Gs coupling only occurs if the adenylyl cyclase is stimulated receptor-independently by forskolin (or if Gs itself is overexpressed). If another GPCR (beta2 receptor) is stimulated, only the already known Gi coupling of the MT1 receptor was noted. This makes the relevance of the findings somewhat questionable.

Taken together, it was shown that the human MT1 receptor theoretically can couple to both Gi and Gs proteins. But the biological relevance still seems questionable. These points have to be discussed in more detail.

Minor:

In the Methods, the origin of all used cDNAs should be given, including gene bank numbers.

In the legend to Figure 1, the abbreviation for melatonin has been swapped with that of pertussis toxin.

Page 8, line 8: it should probably say: …, with the maximum of more than …..

Please indicate the number of experiments in all figure legends (of course not for Figure 7).

Reviewer 2 Report

Comments and Suggestions for Authors

The manuscript entitled Real-time determination of intracellular cAMP reveals functional coupling of Gs protein to the melatonin MT1 receptor is presented for the peer review. Melatonin is a neuroendocrine hormone that regulates the circadian rhythm and numerous another physiological processes. Its functions are primarily exerted through two subtypes of human melatonin receptors, termed melatonin type-1 (MT1) and type-2 (MT2) receptors. Both MT1 and MT2 15 receptors are generally classified as Gi-coupled receptors owing to their well-recognized ability to 16 inhibit cAMP accumulation in cells.GPCRs bind a tremendous variety of signaling molecules, yet they share a common architecture that has been conserved over the course of evolution. Many present-day eukaryotes — including animals, plants, fungi, and protozoa — rely on these receptors to receive information from their environment. 

Manuscript is well-written and structured. It makes good impression but Ihave several notes to authors to improve the overall quality.

1. One biochemical method is not enough to draw such conclusion. I recommend on making additional experiments on loss-of -function or gain-of function at least in vitro to prove your conclusions. For example, apply siRNA specifically blocking MT1 or MT2 expression inducing temporal knockout of gene of interest.

2. This phenomenon is described only in HEK293. Could you repeat it in another cell lines?

3.   Have you tested PEI for cytotoxicity. Please provide phrase in Methods and mention this in Results chapters. 

4. Also provide information on time of incubation before GloSensor cAMP assay. It is not clear from your description in Methods part. How long the incubation lasted? Have time of incubation affected the received data on cAMP?

5. Please provide statistical difference in Figure, especially in Fig.1

6.Overexpression of Overexpression of Gαs sensitized the forskolin-induced cAMP production (Figs. 5B vs 5A). Have you downregulate Gas to prove forskolin impact?

lo

Comments on the Quality of English Language

English is OK.

Reviewer 3 Report

Comments and Suggestions for Authors

Tse et al., investigated effects of melatonin application to HEK293 cells expressing melatonin GPCRs using a highly sensitive real-time luciferase-based approach. Given the physiological role of melatonin, the study is undoubtedly important. The research is significant and scientifically sounds well. Authors propose a model in which melatonin receptors exert opposite signaling depending on availability of Gs or Gi proteins. However, there are minor issues which should be corrected:

1.  Numbers of independent experiments should be clearly indicated in the methods or in the legends of every figure if they are different. 

2.  How mean was calculated should be clearly written. Was it mean for different wells or for independent experiments?

3. Statistical tests used for estimating significance should be clearly indicated.

4.  In line 90, time resolved fluorescence sensors using genetically encoded cAMP sensors require similar efforts and identical workflow as for GloSensor. For example, see  [1] where different sensors were compared. The choice of GloSensor cannot be justified by superiority over them in terms of less required incubations.

Comments on the Quality of English Language

In line 132, word “flexibility” instead of “instability” would better explain why the IL3 is not resolved in crystal structures

Round 2

Reviewer 1 Report

Comments and Suggestions for Authors

The authors have responded adequately to all my concerns.

Reviewer 2 Report

Comments and Suggestions for Authors

Authors have cleared all my issues.